# Long arcuate fascicle in wild and captive chimpanzees as a potential structural precursor of the language network

Yannick Becker [1] ✉, Cornelius Eichner [1], Michael Paquette[1], Christian Bock [2], Cédric Girard-Buttoz[3,4], Carsten Jäger [5,6], Tobias Gräßle[7], Tobias Deschner[8,9,10], EBC Consortium*, Philipp Gunz [11], Roman M. Wittig [3,12,13], Catherine Crockford[3,12,13], Angela D. Friederici [1] & Alfred Anwander [1]

The arcuate fascicle (AF) is the main fibre tract in the brain for human language. It connects frontal and temporal language areas in the superior and middle temporal gyrus (MTG). The AF's connection to the MTG was considered unique to humans and has influenced theories of the evolution of language. Here, using high-resolution diffusion MRI of post-mortem brains, we demonstrate that both wild and captive chimpanzees have a direct AF connection into the MTG, albeit weaker than in humans. This finding challenges the notion of a strictly human-specific AF morphology and suggests that language-related neural specialisation in humans likely evolved through gradual evolutionary strengthening of a pre-existing connection, rather than arising de novo. It is likely that this neural architecture supporting complex communication was already present in the last common ancestor of hominins and chimpanzees 7 million years ago, enabling the evolution of language processes in the human lineage.

Language defines the human species, but the evolution of its neural basis is still largely unknown. In the human brain, the main fibre tract connecting language-relevant regions is the left-lateralised arcuate fascicle (AF), which connects temporal and frontal language areas[1,2]. In the temporal lobe two distinct connections exist: While auditory-motor mapping (repetition) is assured by the superior temporal (STG) connection, lexical-semantic and syntactic mapping (phrase generation) is supported by the strong and lateralised middle temporal (MTG) connection[1–4]. It has been hypothesized that a major morphological transformation occurred in human AF white matter, accounting for our specificity in language processing[5]. Understanding human-specific AF specialisations over the course of evolution is therefore fundamental for theories on language evolution[2,4,5].

Studying the AF evolution, through comparative studies between primate species, can shed light on the evolution of language. The consensus in the comparative literature is that monkeys and apes show

[1]Department of Neuropsychology, Max Planck Institute for Human Cognitive and Brain Sciences, Leipzig, Germany. [2]Alfred Wegener Institute, Helmholtz Centre for Polar and Marine Research, Bremerhaven, Germany. [3]Evolution of Brain Connectivity Project, Max Planck Institute for Evolutionary Anthropology, Leipzig, Germany. [4]ENES Bioacoustics Research Lab, CNRL, University of Saint-Etienne, CNRS, Inserm, Saint-Etienne, France. [5]Department of Neurophysics, Max Planck Institute for Human Cognitive and Brain Sciences, Leipzig, Germany. [6]Paul Flechsig Institute – Center of Neuropathology and Brain Research, Medical Faculty, University of Leipzig, Leipzig, Germany. [7]Ecology and Emergence of Zoonotic Diseases, Helmholtz Institute for One Health, Helmholtz Centre for Infection Research, Greifswald, Germany. [8]Max Planck Institute for Evolutionary Anthropology, Leipzig, Germany. [9]Comparative BioCognition, Institute of Cognitive Science, Osnabrück University, Osnabrück, Germany. [10]Ozouga Chimpanzee Project, Loango National Park, Loango, Gabon. [11]Department of Human Origins, Max Planck Institute for Evolutionary Anthropology, Leipzig, Germany. [12]Institute for Cognitive Sciences Marc Jeannerod, UMR CNRS 5229, University Claude Bernard Lyon 1, Bron, France. [13]Taï Chimpanzee Project, CSRS, Abidjan, Ivory Coast. *A list of authors and their affiliations appears at the end of the paper. ✉e-mail: beckery@cbs.mpg.de

the auditory-motor connectivity but not the core language network connectivity. In other words, nonhuman primates are thought to have an un-lateralised AF that connects the regions homologous to human language areas in the frontal cortex to areas in the superior temporal gyrus, but not to the middle temporal gyrus[6–13]. It has been proposed that the evolution of this human unique direct AF connection with the mid-temporal areas cannot be attributed solely to cortical expansion[9–11]. If the AF-MTG connection is missing in monkeys and apes, lexical-semantic and syntactic mapping through the AF could not be achieved[4,14,15].

Previous work described an MTG-targeted connection in one out of four chimpanzees[14], and left-lateralisation of the STG connection at the species level[15]. However, subsequent studies using the same database of captive individuals have not observed these traits (AF-MTG connection[8,10–12], left lateralisation[11,12,16]), leading to the consensus that both traits are unique to humans. Here, we re-examine chimpanzee AF anatomy using independent high-resolution data from the Evolution of Brain Connectivity Project (EBC)[17,18], which includes data from both wild and captive individuals.

Our sample comprises brains from chimpanzees who died naturally or unavoidably, collected from African wildlife field-sites, sanctuaries, and European zoos[17,18] ($N = 20$ individuals, 39 hemispheres: 20 left, 19 right; 8 females, 12 males; all adult or sub-adult with age >10 y). We optimised diffusion MRI acquisition for post-mortem measurements (55 diffusion directions, optimised diffusion weighting[19], b = 5000 s/mm$^2$, on a preclinical high-field MRI system, resulting in high-quality data with ultra-high spatial and angular resolution (for details see[20] and Methods). The isotropic resolution of 500 μm is up to 46 times more detailed than previously analysed chimpanzee data, allowing the AF to be studied with exceptional precision (Fig. 1A). For comparison, we analysed high-resolution diffusion MRI data from 20 sex-matched healthy human participants (8 females, 12 males).

## Results

### Deterministic tractography
After visually identifying AF white matter structures on colour-coded images (see Fig. 1A), deterministic arcuate fascicle tracking was conducted in these high-resolution data by virtually dissecting whole brain tractograms in the individual space. Regions in the inferior frontal gyrus and middle temporal gyrus were used as seed regions, along with an exclusion mask in the ventral insula. Results revealed high variability in temporal connectivity in chimpanzees. Notably, a connection with the MTG was identified in 9 out of 39 hemispheres (see Fig. 1C) (left hemispheres: 5 out of 20, right hemispheres: 4 out of 19; in total 8 of 20 individuals) using this dissection technique.

### Quantitative probabilistic tractography
To enable direct comparison with previous studies[8–15,16], we then performed FSL standardised observer-independent probabilistic tractography[21] using a standardised chimpanzee template[22]. Anatomically defined temporal ROIs from the DAVI130 atlas[22] with adaptations of the posterior boundary[12] and tractography ROIs[8] included: a parietal seed ROI in the core of the AF, a waypoint mask in the frontal lobe, and either the posterior MTG as a second waypoint mask for a possible AF-MTG connection, or the posterior STG as a second waypoint mask for the AF-STG connection (Supplementary Figs. S2, S3, all masks available as Supplementary files). The ROIs were morphed to the individual brain and the connectivity strength (the number of streamlines connecting the temporal and the frontal ROI) were computed and normalised by the size of the individual seed mask. Remarkably, a direct AF-MTG connection was detected in all 20 individuals using probabilistic tractography. In 2 individual hemispheres, the AF-MTG connection was not detected with sufficient robustness using probabilistic tractography (see Methods and Fig. 2B) and the subjects were excluded from the quantitative lateralisation analysis.

For validation with a more conservative method, systematic deterministic tractography was also performed using the same ROIs, and the results showed the same connectivity patterns as the systematic probabilistic tractography (see Supplementary Figs. S4 and S5 for a comparison of the results).

### Comparison of connectivity strength between humans and chimpanzees
Next, connectivity strength and lateralisation were compared between the AF-STG and the AF-MTG connection in chimpanzees and humans using a Bayesian linear mixed model (see Methods). In chimpanzees, we found strong statistical support (supported by 100% of the posterior) for the AF-STG connection to be stronger than the AF-MTG (median connectivity 14.3- and 45.3- times stronger for the left and right hemispheres, respectively) (Fig. 2C). Notably, we found strong support (100% of the posterior distribution) for a reversed ratio of AF-STG to AF-MTG connectivity in humans (median connectivity 6.32- and 2.5-times stronger AF-MTG than AF-STG connectivity for the left and right hemispheres, respectively).

### Individual-level hemispheric specialisation
Regarding individual-level hemispheric specialisation, we computed an Asymmetry Quotient (AQ) for the right (R) and left (L) hemispheres ($AQ = (R - L)/[(R + L) \times 0.5]$), classifying individuals as left-lateralised: $AQ \leq -0.025$, right-lateralised: $AQ \geq 0.025$, or unbiased: $-0.025 < AQ < 0.025$ (Fig. 3). Individuals with robust results (e.g. >50 streamlines) and for whom both hemispheres were available were included. In the case of chimpanzees, lateralisation was found in all individuals for AF-MTG connections and all but one individual for the AF-STG connection. The AF-STG was left-lateralised in 10 out of 19 individuals and right-lateralised in 8, while the AF-MTG was left-lateralised in 11 individuals out of 17, and right-lateralised in 6 (Fig. 3). In contrast, 15 out of 20 human participants exhibited left lateralisation for the AF-STG and 19 showed left-lateralisation for the AF-MTG.

### Group-level hemispheric specialisation
Regarding group-level hemispheric specialisation in chimpanzees the left AF-STG connection is 35% stronger than the right, whereas the left AF-MTG connection is 4.3 times stronger than the right. In humans the left AF-STG connection is 26% times stronger than the right, whereas the left AF-MTG connection is 3.18 times stronger than the right (Median connectivity, Fig. 2C and Supplementary Fig. S8).

Our statistical model confirmed the known overall strong left lateralisation in humans (99.3% posterior support) with some support (93.6% posterior support) for AF-MTG to be more lateralised than AF-STG in humans and chimpanzees. We found no support for the three-way interaction species*hemisphere*tract type (only 56.6% posterior support) nor for the two-way interaction species*hemisphere (only 51.2% posterior support). This indicates that, albeit being numerically weaker, the left lateralisation in chimpanzee tracts is not statistically different from the human left lateralisation. It also indicates that in chimpanzees, as in humans, AF-MTG tends to be more lateralised than AF-STG.

### Wild versus zoo comparisons
Regarding differences in brain connectivity associated with living environments in the wild or in the zoo, the small sample size did not allow statistical testing of group differences. However, a possible plasticity for the AF-MTG pathway related to the living environment should be noted. 8 out of 9 zoo-housed chimpanzees show a left lateralisation of this connection, whereas wild individuals show a more diverse pattern (2 individuals are left lateralised and 4 are right lateralised) (Fig. 3).

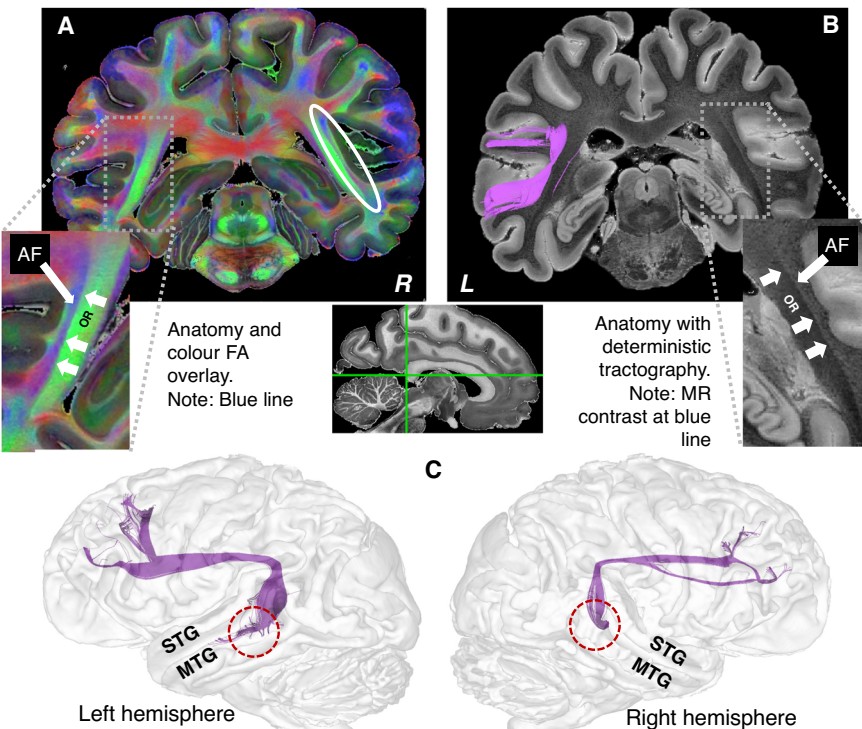

**Fig. 1 | Descriptive AF-MTG evidence. A** High resolution colour FA image overlay on an anatomical image of one exemplary chimpanzee. Note the descending blue line representing white matter fibres running in top-down orientation, lateral to the optic radiation (green). The absence of this blue line has been reported in previous chimpanzee studies, but it is present in humans[15]. **B** Deterministic tractography (purple) overlay on an anatomical image demonstrating the descending arcuate fascicle (AF-)MTG connection at the location from the blue line in the colour FA image (see (**A**)). Also, this structural image shows a clear contrast between the optic radiation and the lateral structure representing the AF. OR Optic radiation. **C** Example individual in sagittal view depicting a long AF-MTG connection as assessed by virtual dissection deterministic tractography. STG superior temporal gyrus, MTG middle temporal gyrus.

## Discussion

Our data, which are based on high-resolution brain scans, demonstrate an emerging direct lateralised AF-MTG connection in both wild and captive chimpanzees. This is a stark contrast to catarrhine monkeys, in which a middle temporal gyrus cannot be distinguished from an inferior temporal gyrus (Fig. 4). Consistent with previous research[7,9], our post-mortem high-resolution tractography of a macaque revealed no connections extending beyond the STG into the lower temporal lobe (see also Supplementary Fig. S1 and ref. 19).

A likely reason this finding has only now been observed is the high resolution and quality of this new dataset. Notably, the strength of the AF-MTG connection is much weaker than the AF-STG connection, a ratio that is reversed in humans[1]. Our data refines the previous model[9,10,14] by adding a continuous aspect to the massive increase of the AF-MTG connectivity, in parallel with language evolution in the human lineage.

Our findings on hemispheric lateralisation support this gradual evolutionary perspective. Although only a weak AF-MTG population-level left asymmetry was observed, in contrast to strongly left lateralised humans[9,16], all individuals showed lateralisation, with a majority displaying left-sided dominance. Most of these left-sided individuals are zoo-housed and might drive the group-level left lateralisation.

Previous data suggest that strong interaction with humans and "do-as-I-do" imitation training resulted in leftward shifts in the fronto-temporal pathway in captive chimpanzees[23]. Consequently, due to special training and animal selection zoo-housed individuals may not be fully representative of the wild chimpanzee populations[24]. The sample size did not permit statistical testing of group differences in connectivity or lateralisation. However, our descriptive data indicate a potential brain plasticity effect on AF-MTG lateralisation influenced by living conditions. All but one of the zoo-housed chimpanzees were left-lateralised, while the wild group exhibited a more varied pattern (Fig. 3).

Future studies are needed to investigate possible structure-function relationships of AF anatomy in chimpanzees. The regions connected by the AF-STG pathway, have been shown to be related to intentional communication in chimpanzees and baboons[25]. However, the behavioural correlates of this long AF connecting the MTG in chimpanzees are currently unknown. Interestingly, chimpanzees' goal-directed vocalisations and gestures suggest possible precursors of syntax and semantics, with some having specific intentional meanings[26–28], and with some gestures exchanged in face-to-face turn-taking with a similar latency to that of human conversation[29]. In contrast to catarrhine monkey vocal repertoires, the vocal utterances of chimpanzees contain structural properties where units are either emitted singly or are routinely and flexibly combined into longer utterances[30], some of which are compositional[31], likely conveying several meanings within a single utterance[32] and may result from vocal usage learning[33,34]). The long AF, which connects the MTG with the prefrontal cortex, including the homologue of Broca's area[35], may therefore represent the neural architecture underlying complex communication of chimpanzees, with some goal-directed produced gestures[36,37] and vocalisations[27,28,33,38].

Our findings suggest that the human-specific AF morphology likely evolved by strengthening an existing connection in the brain rather than arising de novo. It is likely that this neural scaffolding was already present in the last common ancestor of hominins and chimpanzees, and enabled the evolution of language processes in the human lineage.

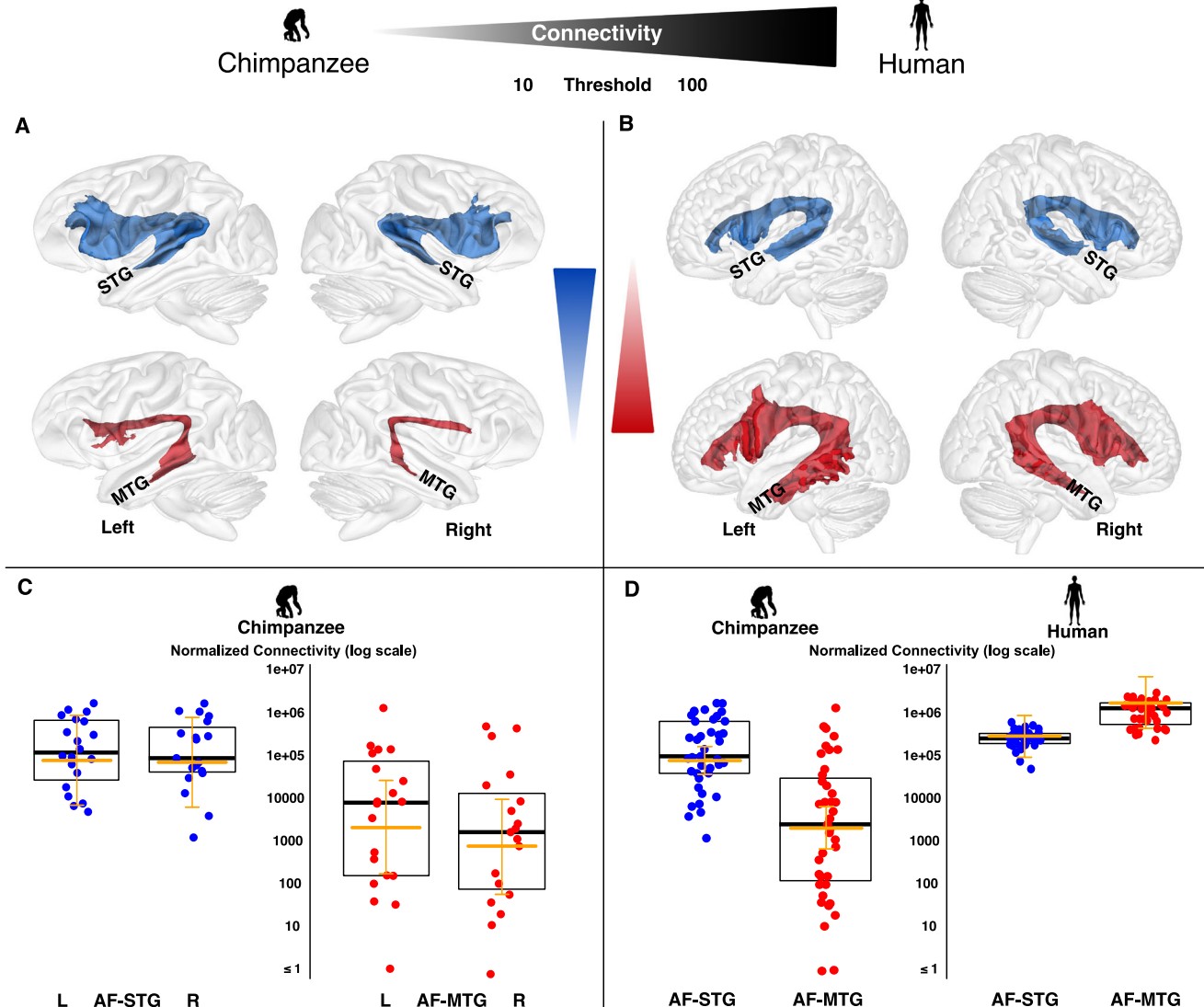

**Fig. 2 | Quantitative MTG connectivity of the arcuate fascicle (AF).** Group average of the probabilistic AF-STG (blue) and AF-MTG (red) tractography results for the right and left hemispheres and for chimpanzees (**A**) and humans (**B**). Average tractography shows the pathway at more than 10 probabilistic streamlines per voxel for chimpanzees (**A**) and more than 100 streamlines per voxel for humans (**B**). For different thresholds see Supplementary Fig. S6. **C** Individual normalised connectivity strength for the AF-STG (blue) and AF-MTG (red) on a log scale for the left hemispheres ($N=20$) and right hemispheres ($N=19$). **D** Comparison of AF-STG and AF-MTG connection strength between humans and chimpanzees, highlighting an inverse pattern of strength. In chimpanzees, median AF-STG connection was found to be 14.3- (left, $N=20$) and 45.3- (right, $N=19$) times stronger than AF-MTG pathway for the left and right hemispheres, respectively. In humans median AF-MTG connection was found to be 6.32- (left, $N=20$) and 2.5- (right, $N=20$) times stronger than the AF-STG pathway. The boxplots show the median (black thick line), the 25% and 75% quartiles and the individual data points. The orange horizontal lines represent the model mean estimates and the 95% Credible Interval (CI). Comparison of AF-MTG lateralisation between chimpanzees and humans is shown in Supplementary Fig. S8. Individual tractography results are shown in Supplementary Fig. S5 and are provided as supplementary information files. STG superior temporal gyrus, MTG middle temporal gyrus.

## Methods

### Chimpanzee dataset

**Population.** 20 post-mortem (PM) chimpanzee (*Pan troglodytes*) brains (39 hemispheres: 20 left, 19 right; 8 females, 12 males; age range 10–61 years, age mean: 34.95, stdev: 16.2) were analysed in this study. For brains from field sites the post-mortem interval before fixation (PMI) was $12 \pm 8$ h and the mean PMI for brains from sanctuaries and zoos was $9 \pm 7$ h. Conservation of the brains was done in 4% paraformaldehyde in phosphate buffered saline (PBS) pH 7.4 for at least 8 weeks. Tissue quality, integrity, and fixation were good and of a high standard for PM diffusion MRI,

as assessed by fixation- MRI- and histology screenings (for more information see Grässle et al.[18]).

**Chimpanzee scanning procedure.** The brain was scanned in a spherical container with perfluoropolyether ("Fomblin®" PFPE) sealed using additional synthetic foil packaging. Comprehensive padding with sponges was applied to both the brain in the sample container and the sample itself to reduce mechanical coupling between the specimen and the MRI system during data acquisition[17].

We acquired ultra-high resolution diffusion magnetic resonance images (dMRI) on a Bruker Biospec 94/30 at 9.4 T MRI scanner (Bruker

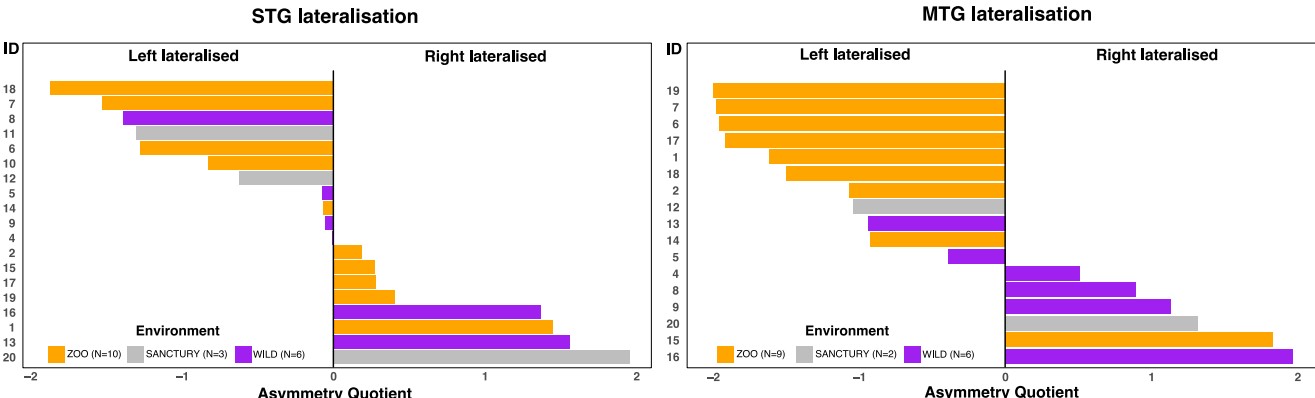

**Fig. 3 | Asymmetry quotient for individual chimpanzee brains for the AF-STG (right panel) and AF-MTG (left panel) connections.** Left: For the AF-STG, all but one individual are lateralised (19 individuals meet the inclusion criteria) and 10 (53%) chimpanzees are left lateralised whereas 8 (42%) are right lateralised. For the AF-MTG connection, 11 (65%) chimpanzees are left lateralised, and 6 (35%) are right lateralised. Wild individuals (purple), zoo-housed individuals (orange), sanctuary-housed individuals (grey). The corresponding individual connectivity values are shown in the Supplementary Fig. S9. AF arcuate fascicle, STG superior temporal gyrus, MTG middle temporal gyrus.

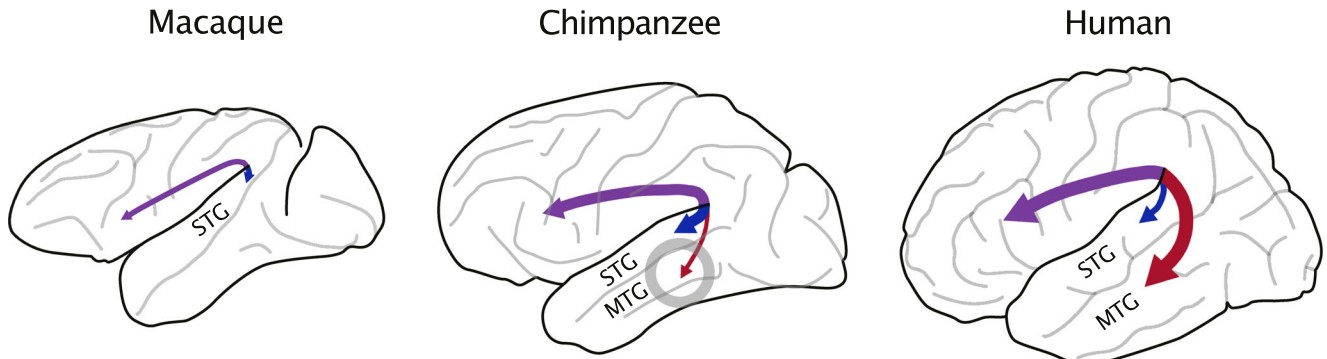

**Fig. 4 | Schematic summary of the findings on the temporal endings of the AF (purple).** The figure shows the AF-STG (blue) and AF-MTG (red) in the left hemisphere of macaques, chimpanzees, and humans (Supplementary Fig. S1). Our results replicate the AF-STG connection in chimpanzee[8,11,14,15,19] and macaques[7,9,19] (Fig. 2; Supplementary Fig. S1); but highlight a human-like AF-MTG connection in the chimpanzees, although weaker than in humans (Fig. 2). AF arcuate fascicle, STG superior temporal gyrus, MTG middle temporal gyrus.

BioSpin, Ettlingen, Germany with Paravision 6.0.1) using a strong gradient system (BGA 20, 300 mT/m) and a 3D segmented spin-echo EPI sequence[19] (500 μm isotropic resolution, b-value = 5000 s/mm² in 56 directions and 3 b0 images, TE = 58.9 ms, TR = 1000 ms, matrix: 240*192*144, EPI segmentation factor 32, one b0 with opposite phase encoding), providing data of unprecedented quality[20]. In addition, a noise map was acquired to debase the diffusion data with matching parameters. Finally, 5 structural FLASH images were acquired with variable flip angles and matching resolution for the structural segmentation of the brain tissues. The total acquisition time was about 90 h per brain[20].

### Chimpanzee dMRI analyses

**Preprocessing.** dMRI data was preprocessed using a pipeline optimised for PM data[20]. This included signal debiasing, denoising, 3D Gibbs ringing correction, field-drift correction, distortion correction, motion, and eddy current correction using the implementation in FSL (v6.0) (http://www.fmrib.ox.ac.uk/fsl) and MRtrix3 (v3.0.2) (https://www.mrtrix.org). The structural images were segmented into grey matter and white matter using a multi-contrast fuzzy c-means clustering[20].

**Deterministic tractography for virtual dissections.** The initial deterministic tractography used a novel model of crossing fibres. This

model was derived from the Constrained Spherical Deconvolution (CSD) model to account for variable tissue properties across the brain and is adapted to PM data. This new Local Spherical Deconvolution (LSD) reconstructs fibre orientation density functions (fODFs) using a voxel-specific optimal deconvolution kernel adapted to local tissue properties. It accounts for different microstructural environments surrounding the axons and best represents the acquired dMRI data (for more details see Eichner et al.[20]). This local model was used to compute whole-brain deterministic streamline tractography implemented in MRtrix3. A streamline was initiated in each white matter voxel using the following parameters: step size of 0.125 mm, minimum and maximum length of 10 mm and 200 mm, respectively, and a fODF cut-off of 0.1. Finally, we virtually dissected the component of the arcuate fascicle (AF) connecting the middle temporal gyrus (MTG) in each hemisphere. This was achieved using a frontal region in the inferior frontal gyrus and a temporal region in the posterior MTG, together with an exclusion mask in the ventral insula and adjacent white matter.

**Observer independent systematic deterministic tractography.** For systematic observer independent deterministic tractography, we used the same local ODF fibre crossing and tracking model as for virtual dissection. We seeded 100 streamlines per voxel in a conservatively

selected large white matter region containing all potential branches of the arcuate fascicle. Tractography was computed in a full white matter mask using the following parameters: relative ODF threshold: 0.1, step size: 0.5 mm, maximum streamline length: 100 mm, angular threshold: 85°, tracking in both directions. Streamlines for the AF-STG and AF-MTG connection were selected using the same frontal, parietal and temporal ROIs as in the probabilistic tractography model. To account for potential partial volume effects in streamline selection, all selection ROIs were dilated by 0.5 mm.

**ROI definition for probabilistic tractography.** For refined probabilistic tractography, we defined anatomical frontal and temporal regions-of-interest (ROIs) in a chimpanzee template (JUNA) by adapting a previously published atlas (DAVI130[22]). In particular, the posterior boundaries of the temporal areas were corrected[12]. We used a parietal white matter seed ROI in the core of the AF[8] and waypoint masks in the frontal lobe and either the posterior MTG or the posterior STG as a second waypoint mask for the AF-STG connection. The ROIs in the temporal lobe were conservatively defined to include only the lateral (upper) half of the gyrus (including grey and white matter) to avoid false-positive connections not reaching into the gyrus. Therefore, no conclusions can be made for the region between the STG and MTG, which is the STS. The frontal mask included the inferior and middle frontal gyrus (IFG/MFG) to capture the full frontal fanning of the AF. In addition, exclusion masks were defined in the insula, the extreme capsule, in the interhemispheric plane[8] in the MTG for AF-STG tractography and in the STG for AF-MTG tractography. For the AF-MTG tractography, we cannot not exclude that the AF is also reaching to other more ventral areas.

**Template registration.** The chimpanzee template (JUNA, 0.5 mm) was non-linearly registered to the individual brain using the SyN method of ANTs (v2.3.5) (https://picsl.upenn.edu/software/ants). Registration was performed using a multi-contrast image, i.e. using all 5 acquired FLASH images equally within the mutual information metric. The registration was performed using 5 resolution levels to account for the high image resolution and with SyN parameters optimised for the task (notably gradientStep = 0.3 and updateFieldVarianceInVoxelSpace = 7.5). Using this deformation field, the ROIs were morphed to the individual brains.

**Probabilistic tractography.** To quantify connectivity differences between hemispheres and brains and to ensure comparability with previous studies, we performed standardised observer-independent probabilistic tractography[21]. Therefore, we seeded 20,000 streamlines per voxel in the parietal seed mask (parameters: curvature threshold 0.2, samples per voxel of 2000, step length of 0.5). The streamline was selected if it crossed both, the temporal and the frontal mask, resulting in a continuous tracking of the long segment of the AF[39]. To control for individual differences in the size of the seed region, we divided the number of AF streamlines by the size of the seed region (voxels) and multiplied the values by 1000. This number corresponds to the average size of the white matter seed mask. To exclude false-positive connections, we classified all tracts with probabilistic connectivity values below 50 (sum of both hemispheres) as 'not robust' for lateralisation analysis. Above this threshold, probabilistic tractography allows high confidence in the existence of the connection. As an additional categorisation, we classified individual tracts with connectivity values below 1000 as weak connections and above 1000 as strong connections (see Supplementary Fig. S.5).

**Visualization.** To reduce the dynamic range of the connectivity values and to obtain a more normal distribution, we log-transformed the 3D tractography map (number of streamlines per voxel). Individual tractography maps were then warped into template space, and averaged to produce group overlap maps. Both, the mean AF-STG connection and the AF-MTG connection were thresholded at a value of 10 streamlines per voxel, for optimal comparability and visualisation of the shape of the pathway.

**Human probabilistic tractography**
In order to compare the chimpanzee results with human data, the same probabilistic tractography settings and the same inclusion and exclusion ROIs were used in both species.

High-resolution diffusion MRI images of 20 young healthy adult participants (8 females, sex-matched; age range 27–33 years, age mean: 29.05, stdev: 1.88) were randomly selected from a larger cohort of previously measured data. We acquired high-resolution dMRI data on a Siements 3T Prisma MRI system (Siemens Healthineers, Erlangen, Germany) using a 32-channel head coil (1.3 mm isotropic resolution, b-value = 1000 s/mm² in 60 directions and 7 b0 images, TE = 75 ms, TR = 6 s, CMRR-SMS = 2, GRAPPA = 2, 3 repetitions, and 2 b0 images with opposite phase encoding directions). In addition, a structural image with 1 mm resolution was acquired as anatomical reference.

Pre-processing followed an optimised pipeline[40,41], which included motion, eddy current and distortion correction, as well as tensor fitting and estimation of a probabilistic fibre crossing model using FSL (v6.0). The masked FA images of the brains were non-linearly registered to the MNI space represented by the FSL_HCP1065_FA human template brain using the SyN method of ANTs (v2.3.5). This template was also used to define the frontal, temporal[42] and parietal ROIs for tractography as well as the exclusion masks (sagittal, medial to the AF and coronal in the extreme capsule and in the posterior insula, see Supplementary Fig. S7). The computed registration fields were used to morph the ROIs to the individual brains and probabilistic tractography was performed in the white matter mask following the same procedure as for the chimpanzees. The AF-STG and AF-MTG connectivity values and the probabilistic tractography maps were scaled to a reference seed size of 1000 voxels. For visualization, the maps were log-transformed and spatially normalised to the reference MNI space using the same procedure as for chimpanzees to generate the group-averaged tractography results.

**Macaque dMRI tractography**
An ultra-high resolution dMRI PM macaque dataset was analysed in a previous publication of the group[16] from an openly available resource[43]. For comparison, we have included the results of probabilistic dMRI tractography of the AF[16] in Supplementary Fig. S1.

**Statistics**
We ran a Bayesian generalised linear mixed model with a lognormal error distribution to assess the effect of fibre tract type (AF-STG or AF-MTG), brain side (left or right) and species (chimpanzees or humans) on connectivity strength (the number of streamlines connecting the temporal and the frontal ROI). In this model, we controlled for the sex and the age of the individuals as fixed factor. We also included individual identity as random effect to account for repeated measures in the same brain (left/right hemisphere and AF-STG and AF-MTG). The conditional R square indicating the variance explained by the random and fixed effect was 0.86 and the marginal R square indicating the variance explained only by the fixed effects was 0.61. The difference between the conditional and marginal R squares indicates a strong contribution of inter-individual differences in explaining variance in AF-STG and AF-MTG strength.

In all statistical models, age was scaled to a mean of 0 and an SD of 1. For the different plots, we reran each model with a centred factor sex. This allowed to obtain the model estimate for mean tract strength in each tract type, in each hemisphere and in each species for an average individual.

All statistical models were done using the function brm from the brms package (version 2.19.0, https://paul-buerkner.github.io/brms) in R (version 4.3.1).

### Ethics statement

The animal study was approved by the National Research and Animal Welfare Authorities in the country of origin, as well as by the Ethics Committee of the Max Planck Society for Field Research. The research adheres to the IUCN best practice guidelines for health monitoring and disease control in great ape populations. Brains were extracted postmortem from individuals that died naturally. In rare instances, brains were obtained from European zoos when individuals were euthanized due to terminal medical conditions, or from the wild following human-animal conflicts. None of these events were controlled by anyone involved in the research. Since our procedure was non-harmful to the animals, it is considered non-invasive from an ethical standpoint. The study was conducted in accordance with local legislation and institutional requirements. Human participants gave informed consent, and the acquisition of the human data was approved by ethics committee of the University of Leipzig (President Prof. Dr. R. Preiss).

### Reporting summary

Further information on research design is available in the Nature Portfolio Reporting Summary linked to this article.

## Data availability

The individual tractography data generated in this study have been deposited on the server of the Max Planck Digital Library available under this link (https://openscience.cbs.mpdl.mpg.de/ebc/af-mtg_tractography/tractography_data.zip).

## Code availability

Code and processing routines for diffusion MRI analyses were previously published and are publicly available for download at: https://github.com/mpaquette/EBC_dMRI_Preprocessing/tree/fullproc. Code for statistics and figures in this study is provided in a dedicated GitHub repository: https://github.com/tozbu/AF_chimp_brain.

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

## Acknowledgements

We are thankful for the permissions granted for research and brain extraction in various regions: Sierra Leone's Environmental Protection Agency; the Ministère de l'Enseignement Supérieur et de la Recherche Scientifique and the Ministère de Eaux et Fôrets in Côte d'Ivoire, along with the Office Ivoirien des Parcs et Réserves; in Gabon, the Ministère des Eaux, des Forêts, de la Mer, de l'Environnement, the Agence Nationale des Parcs Nationaux, the Research Institute for Tropical Ecology, and the National Center for Scientific and Technological Research; and in Uganda, the Uganda Wildlife Authority and the Ugandan National Council for Science and Technology. Our appreciation also goes to the Centre Suisse de Recherches Scientifiques en Côte d'Ivoire for their logistical support, the staff members of the Taï Chimpanzee Project for their invaluable assistance in data collection, and to Christophe Boesch for his exceptional dedication to establishing and nurturing the Taï Chimpanzee Project for 30 years. We express our gratitude to all members of EAZA, PASA, and field site partners who collaborated with us on the Evolution of Brain Connectivity (EBC) project. Special acknowledgment to Christina Kompo for her adept management of import and export permits and shipping logistics. This study received funding from the Max Planck Society under the inter-institutional funds of the president of the Max Planck Society for the EBC project. Y.B. received funding from the Fondation Fyssen.

## Author contributions

Designed research: Y.B., A.D.F., R.W., C.C., C.E., A.A. Analysed data: Y.B., C.E., M.P., C.G-B., A.A. Wrote the paper: Y.B., C.G-B, P.G., C.C., A.D.F., A.A. Acquired MRI data: Y.B., M.P., C.E., C.B., A.A. Acquired brain specimens: EBC Consortium; T.G., T.D., C.J., R.W., C.C.

## Funding

## Competing interests

The authors declare no competing interests.

## Additional information

## EBC Consortium

Bala Amarasekaran[14], Alfred Anwander [1], Caroline Asiimwe[15], Daniel Aschoff[16], Yannick Becker [1] ✉, Martina Bleyer[16], Christian Bock [2], Julian Chantrey[17], Catherine Crockford[3,12,13], Tobias Deschner[8,9,10], Cornelius Eichner [1], Pawel Fedurek[18], Karina Flores[14], Angela D. Friederici [1], Cédric Girard-Buttoz[3,4], Zoro Bertain Gone Bi[19], Philipp Gunz [11], Tobias Gräßle[7], Jennifer E. Jaffe[13,20], Carsten Jäger [5,6], Susan Hambrech[21], Daniel Hanus[22], Daniel Haun[22], Evgeniya Kirilina[5], Kathrin Kopp[22], Fabian H. Leendertz[7,13,20], Matyas Liptovszky[23], Patrice Makouloutou-Nzassi[24], Kerstin Mätz-Rensing[16], Richard McElreath[25], Matthew McLennan[26], Zoltan Mezö[27], Sophie Moittié[23], Torsten Møller[28], Markus Morawski[6], Karin Olofsson-Sannö[29], Michael Paquette[1], Simone Pika[9,10], Andrea Pizarro[14], Kamilla Pléh[7,13],

Jessica Rendel[30], Alejandra Romero Forero[14], Jonas Steiner[7,13], Mark F. Stidworthy[31], Lara Southern[9,10], Claudia A. Szentiks[27], Tanguy Tanga[10,24], Reiner Ulrich[32], Steve Unwin[33], Sue Walker[34], Nikolaus Weiskopf[5,35], Gudrun Wibbelt[27], Roman M. Wittig [3,12,13], Kim Wood[36] & Klaus Zuberbühler[15,37]

[14]Tacugama Chimpanzee Sanctuary, Freetown, Sierra Leone. [15]Budongo Conservation Field Station, Masindi, Uganda. [16]Pathology Unit, German Primate Center, Göttingen, Germany. [17]Veterinary Pathology and Preclinical Sciences, School of Veterinary Science, University of Liverpool, Liverpool, UK. [18]Stirling University, Stirling, UK. [19]Centre Suisse de Recherches Scientifiques en Côte d'Ivoire, 01 BP 1303 Abidjan, Côte d'Ivoire & UFR Biosciences, Université Felix Houphouët-Boigny, Abidjan 00225, Côte d'Ivoire. [20]Epidemiology of Highly Pathogenic Microorganisms, Robert Koch Institute, Berlin, Germany. [21]Magdeburg Zoo, Magdeburg, Germany. [22]Department of Comparative Cultural Psychology, Max Planck Institute for Evolutionary Anthropology, Leipzig, Germany. [23]Twycross Zoo, Little Orton, Leicestershire, UK. [24]Institut de Recherche en Ecologie Tropicale, Libreville, Gabon. [25]Department of Human Behavior, Ecology, and Culture, Max Planck Institute for Evolutionary Anthropology, Leipzig, Germany. [26]Oxford Brookes University, Oxford, United Kingdom. [27]Department of Wildlife Diseases, Leibniz Institute for Zoo and Wildlife Research, Berlin, Germany. [28]Kolmarden Wildlife Park, Kolmarden, Sweden. [29]National Veterinary Institute, Uppsala, Sweden. [30]Wuppertal Zoo, Wuppertal, Germany. [31]International Zoo Veterinary Group, Keighley, UK. [32]Institute of Veterinary Pathology, University of Leipzig, Leipzig, Germany. [33]School of Bioscience, University of Birmingham, Birmingham, UK. [34]Chester Zoo, Chester, UK. [35]Faculty of Physics and Earth System Sciences, Felix Bloch Institute for Solid State Physics, University of Leipzig, Leipzig, Germany. [36]Welsh Mountain Zoo, Clauwyn Bay, UK. [37]Institute of Biology, University of Neuchatel, Neuchatel, Switzerland.

