## [Peer Review file · Nature Communications]

Long arcuate fascicle in wild and captive chimpanzees as a potential structural precursor of the language network

Corresponding Author: Dr Yannick Becker

Version 0:

Reviewer comments:

Reviewer #2

(Remarks to the Author)

The revised manuscript studied the existence of the AF-middle temporal gyrus (MTG) connectivity in the brain of the chimpanzees. The study acquired 19 high-quality resolution DTI data from chimpanzees and provides a systematic analysis of the AF-MTG connection in chimpanzees, including lateralization, comparisons with the AF-superior temporal gyrus (STG), and comparisons with the human brain.

Comparing the previous manuscript submitted to Nature Neuroscience, the revised manuscript incorporates several significant improvements:

1. The methods section has been strengthened, confirming the existence of the AF-MTG connection, addressing concerns regarding false positives in tractography, low connectivity probabilities, and offering a more detailed comparison with the human brain.
2. The previously exaggerated interpretation of language evolution (specifically the underlying structure of the AF) in chimpanzees as a "missing link" has been revised to "a possible structural precursor." This revision is more appropriate, although there is little evidence of AF-MTG in previous studies.

In summary, the revised manuscript has addressed the concerns of two reviewers, and introduces some novel aspects and shows improvement. I have no further comments.

Reviewer #4

(Remarks to the Author)

This is an exciting and highly important paper of broad interest in the context of primate brain evolution and especially hominid higher cognitive functions and language abilities. The data are based on a re-examination of postmortem brains from wild and captive chimpanzees of both sexes, and are of exceptional quality, considering the difficulties to access such specimens. The sample is relatively small, but this cannot be considered a weakness considering the species investigated. The methods are well described. The authors have addressed very carefully and comprehensively the original comments of both reviewers, further strengthening their analyses and conclusions. I have no further issues.
